# Osteopontin: A Bone-Derived Protein Involved in Rheumatoid Arthritis and Osteoarthritis Immunopathology

**DOI:** 10.3390/biom13030502

**Published:** 2023-03-09

**Authors:** Beatriz Teresita Martín-Márquez, Flavio Sandoval-García, Fernanda Isadora Corona-Meraz, Erika Aurora Martínez-García, Pedro Ernesto Sánchez-Hernández, Mario Salazar-Páramo, Ana Lilia Fletes-Rayas, Daniel González-Inostroz, Monica Vazquez-Del Mercado

**Affiliations:** 1Departamento de Biología Molecular y Genómica, Instituto de Investigación en Reumatología y del Sistema Músculo Esquelético (IIRSME), Centro Universitario de Ciencias de la Salud, Universidad de Guadalajara, Guadalajara 44340, JAL, Mexico; 2Cuerpo Académico UDG-CA-703 “Inmunología y Reumatología”, Centro Universitario de Ciencias de la Salud, Universidad de Guadalajara, Guadalajara 44340, JAL, Mexico; 3Departamento de Neurociencias, Centro Universitario de Ciencias de la Salud, Universidad de Guadalajara, Guadalajara 44340, JAL, Mexico; 4Departamento de Ciencias Biomédicas, División de Ciencias de la Salud, Centro Universitario de Tonalá, Tonalá 45425, JAL, Mexico; 5Departamento de Fisiología, Centro Universitario de Ciencias de la Salud, Universidad de Guadalajara, Guadalajara 44340, JAL, Mexico; 6Departamento de Enfermería Clínica Aplicada, Centro Universitario de Ciencias de la Salud, Universidad de Guadalajara, Guadalajara 44340, JAL, Mexico; 7Departamento de Biología Molecular y Genómica, Doctorado en Biología Molecular en Medicina, Centro Universitario de Ciencias de la Salud, Universidad de Guadalajara, Guadalajara 44340, JAL, Mexico; 8Hospital Civil de Guadalajara “Dr. Juan I. Menchaca”, Especialidad de Reumatología, Padrón Nacional de Posgrados de Calidad (PNPC) Consejo Nacional de Ciencia y Tecnología (CONACyT), Guadalajara 44340, JAL, Mexico

**Keywords:** osteopontin, rheumatoid arthritis, osteoarthritis, joint and cartilage degeneration

## Abstract

Osteopontin (OPN) is a bone-derived phosphoglycoprotein related to physiological and pathological mechanisms that nowadays has gained relevance due to its role in the immune system response to chronic degenerative diseases, including rheumatoid arthritis (RA) and osteoarthritis (OA). OPN is an extracellular matrix (ECM) glycoprotein that plays a critical role in bone remodeling. Therefore, it is an effector molecule that promotes joint and cartilage destruction observed in clinical studies, in vitro assays, and animal models of RA and OA. Since OPN undergoes multiple modifications, including posttranslational changes, proteolytic cleavage, and binding to a wide range of receptors, the mechanisms by which it produces its effects, in some cases, remain unclear. Although there is strong evidence that OPN contributes significantly to the immunopathology of RA and OA when considering it as a common denominator molecule, some experimental trial results argue for its protective role in rheumatic diseases. Elucidating in detail OPN involvement in bone and cartilage degeneration is of interest to the field of rheumatology. This review aims to provide evidence of the OPN’s multifaceted role in promoting joint and cartilage destruction and propose it as a common denominator of AR and OA immunopathology.

## 1. Introduction

Rheumatoid arthritis (RA) and osteoarthritis (OA) are considered among the major disabling diseases affecting much of the world’s population, resulting in reduced life quality. RA is a systemic chronic autoimmune disease characterized by joint inflammation followed by invasive synovial tissue and destruction of the articular bone and cartilage. OA is a chronic degenerative joint disease typically defined by mechanical abnormalities such as subchondral bone and articular cartilage degradation. In both diseases, the joints are the anatomical site where the pathological process begins; however, there are marked differences between their diagnosis, management, and treatment.

Several proteins have been analyzed as potential candidates to identify new biomarkers that may improve diagnosis and prognosis and could be considered as therapeutic targets. Currently, attention has been drawn to bone-derived proteins. Osteopontin (OPN) is a phosphorylated integrin-binding glycoprotein known to exert an atypical immune regulatory function on chronic inflammatory diseases such as cirrhosis, fibrosis, neuroinflammatory diseases, atherosclerosis, autoimmune diseases, obesity, diabetes, and cancer, among others. Because OPN has been detected in considerably high quantities in RA and OA patients and in vitro analyses and experimental models have proposed it as an effector protein of joint damage, in this review, we propose that OPN could be considered a common denominator of the immunopathology of these rheumatic diseases. 

### 1.1. Osteopontin Gene and Protein Structure

OPN is a phosphoglycoprotein involved in physiological processes such as tissue refurbishing, angiogenesis, bone homeostasis, wound healing, cell adhesion, and immune response [1,2]. The discovery of OPN dates back to the mid-eighties when Franzen and Heingard identified OPN as a sialic acid-rich matricellular protein derived from the bovine bone mineralized matrix [3]. Subsequently, Oldberg and colleagues named this sialoprotein “osteopontin,” a term derived from the Greek word “osteon” (meaning “bone”) plus the Latin word “pons” (meaning “bridge”), which, according to the authors, is the word that better reflects the potential function of OPN as the product of cells in the osteoid matrix [2,4,5,6]. OPN is immobilized in the bone matrix as a component of the extracellular matrix (ECM) and as a soluble protein in human physiological tissues [7,8,9,10]. 

*SPP1* (secreted phosphoprotein-1) is the human OPN encoding gene and is a member of the SIBLING (Small Integrin Binding Ligand N-linked Glycoprotein) family composed of dentin matrix protein 1, bone sialoprotein, dentin sialophosphoprotein, and matrix extracellular phosphoglycoprotein [2,11,12]. The SIBLING proteins, which are located on the human chromosome 4q21, have similarities in exon structures and share an Arg-Gly-Asp (RGD) sequence [12]. *SPP1* is located on 4q21-q25, contains an open reading frame of 942 nucleotides, and gives three messenger ribonucleoprotein (mRNA) variants: OPN-a, OPN-b, and OPN-c. OPN-a contains all exon coding information, OPN-b lacks exon 5, and OPN-c presents a deletion of exon 4 (287 amino acids) [13,14,15]. Nevertheless, it has been observed that the OPN transcript suffers alternative splicing and generates two variants: OPN-4, in which both exons 4 and 5 are deleted, and OPN-5, which presents an additional region derived from exon 3 [16]. 

In gene-related diseases, the mRNA alternative splicing of specific proteins contributes to the pathological process and treatment response. In humans, it has been found that OPN splicing variants play a role in the immunopathogenesis of several kinds of diseases, among them, cancer [17]. Studies have determined that OPN isoforms are disease tissue-specific and exert signaling pathways depending on the availability of ligands produced by the microenvironment. 

Because human studies have limitations, experimental murine models are used to investigate the disease’s pathological mechanisms in detail. In regard to OPN, it is known that murine SIBLING proteins are located on chromosome 5q; its OPN is composed of 294 aa and shows a 59% identity with OPN-a [18]. Although mouse studies had restrictions because no OPN isoforms have been found, recently, Kamalabadi and colleagues detected an OPN splice variant (OPN5) in a murine model of breast cancer [19]. This finding opens the possibility of studying the physiological and pathological role of OPN isoforms in murine models.

### 1.2. Osteopontin Adhesion Motifs and Posttranslational Modifications 

OPN suffers from heavy posttranslational modifications such as phosphorylations, glycosylations, and sulfations; these modifications can be cell type-specific, depend on physiological and pathological factors, and may impact both OPN structure and function [10]. OPN is mainly phosphorylated in Golgi by Family with sequence similarity 20 member C (Fam20C) kinase in humans, which is responsible for OPN phosphorylation in the ECM. OPN phosphorylation regulates its binding interaction with hydroxyapatite (related to bone remodeling) and has been associated with macrophage migration and host–cell interactions [20]. 

OPN is the target of proteolytic modifications, representing a way to regulate its biological activity locally because the properties of cleaved OPN are entirely different from those of the uncleaved protein [21]. Full-length OPN (OPN-FL) contains binding domains related to migration and cell adhesion. When OPN-FL is cleaved by thrombin and matrix metalloproteinases (MMP), it uncovers integrin binding sites and releases a chemotactic C-terminal fragment [9,21,22]. 

One central integrin attachment motif is localized at the ^158^GRGDS^162^ region, which is completely conserved among species, and binds RGD-recognizing integrins α_v_β_1_, α_v_β_3_, α_v_β_5_, and α_8_β_1_, conferring cellular signaling such as endothelial regeneration and cell adhesion and spreading, among others [22,23,24,25,26,27]. Another integrin attachment motif is the ^162^SVVYGLR^168^-containing domain that binds with α_9_β_1_ [28,29,30], α_4_β_1_ expressed by neutrophils and lymphocytes [31,32,33], and α_4_β_7_ [34], allowing adhesion and migration of leukocytes and neutrophils in an RGD-independent manner [9,22,35,36]. 

Contiguous to the RGD domain site at the Arg^168^-Ser^169^ region, a thrombin cleavage site has been identified to expose a new C-terminal sequence and convert OPN-FL in the N-terminal fragment of thrombin-cleaved OPN (OPN N-half). Additionally, two more relevant sequence domains for OPN functionality have been observed and identified at the sequence ^131^ELVTDFTDLPAT^143^, which also binds to α_4_β_1_ [32,37], and the highly conserved sequence ^163^YGLRSKSKKFRR^174^ that protects OPN-FL from cleavage by thrombin [22]. Furthermore, human OPN presents two strong heparin-binding domains at the C-terminal part close to the thrombin cleavage site associated with internalization signals. OPN physiological activity can be modulated by cleavage of MMP and by thrombin-activatable carboxypeptidase B (CBP), which converts OPN-Arg (OPN-R) into OPN-Leu (OPN-L), inactivating the integrin α_9_β_1_ binding site [22,38]. 

OPN’s wide diversity of functions is also related to its capacity to interact with integrins, heparin, calcium, and the cluster of differentiation 44 (CD44) surface receptor [16]. Integrins are heterodimeric transmembrane proteins conformed by non-covalently paired α and β subunits that recognize small peptide sequences such as RGD tripeptides and allow cells to respond to mechanical and chemical properties of the cellular microenvironment [39]. 

Thrombin cleavage changes OPN conformation and allows accessibility to the RGD motif by α_v_β_3_, and SVVYGLR by α_9_β_1_ and α_4_β_1_ integrins in a Ca^+2^-dependent manner [22,40,41]. The major functional receptor for thrombin-cleaved OPN is the α_v_β_3_ integrin, through which binding via a Gly-Arg-Gly-Asp-Ser (GRGDS) motif protects endothelial cells from apoptosis via activation of nuclear factor kappa of activated B cells (NF-kB). Additionally, it contributes to osteoclast (OC) adherence and resorption of bone and haptotaxis of tumor cells, endothelial cells, and vascular smooth muscle cells [42,43]. 

OPN binds with CD44 through the last 18 aa binding domain that is highly conserved at the C terminus [42]. CD44 is a conserved gene generated by alternative splicing and posttranslational modifications of heterogeneous proteins. CD44 presents ten variable exons different from the standard CD44 molecule, the most ubiquitous form expressed by most cell types, including lymphocytes and microglia. CD44 isoforms v3 and v6 are expressed by T cells, where CD44v6 participates in T cell activation and cytotoxic T cell generation. The ligation with CD44 can be done individually or in association with integrin β_1_ in an RGD-independent manner [44]. 

To exert its physiological functions on normal tissues, OPN, through interaction with α_4_β_1_, starts the signaling pathway with degradation of the phosphorylated inhibitor of NF-kB subunit beta (IKKβ), followed by the freeing of the inhibitor of nuclear transcription factor kappa-B alpha (IkBα) and NF-kB (protein 50 and p65 heterodimer). Subsequently, IkBα is degraded, and NF-kB enters the nucleus, is phosphorylated, and enhances the expression of genes related to survival signals. In addition, Fork-head box O3 (FOXO3A) is inactivated by phosphorylated IKKβ, and the signals of anti-survival genes such as B cell leukemia/lymphoma 2 protein (Bcl-2) interacting mediator of cell death (BIM), Bcl-2 homologous antagonist/killer (BAK), and Bcl-2-associated X protein (BAX) are downregulated. 

On the flip side, OPN signaling through α_v_β_3_ and the CD44 RGD domain promotes cell survival, proliferation, tumor progression, and angiogenesis. The later interaction also inhibits complementing by activating the phosphatidylinositol 3-kinase (PI3K)-dependent protein kinase v-akt murine thymoma viral oncogene homolog (Akt) phosphorylation and enhancing the interaction between phosphorylated serine/threonine Akt and IKKα/β [42,45,46,47]. OPN-mediated activator protein 1 (AP-1) acts through the signaling pathway of the nuclear factor-inducing kinase (NIK) extracellular signal-related kinase (ERK) and mitogen-activated protein kinase kinase1 (MEKK1)-c-Jun N-terminal kinase 1 (JNK1) [2,46,47]. Thus, how OPN mediates cellular functions depends on splice variants, exposure to enzymes, and the availability of receptors. Figure 1 represents the human OPN gene, messenger RNA variants, protein structure, and receptors. 

### 1.3. Osteopontin Mediates the Immune Response and Inflammation

OPN, considered a T cell helper 1 (T_H_1) cytokine involved in immune responses, is expressed in dendritic cells (DC), T cells, macrophages, and natural killer (NK) cells. It also works as a pro- and anti-inflammatory molecule that, in certain circumstances, induces the migration of DCs and macrophages to the inflammation’s site, and, in other activities, acts as an anti-inflammatory molecule downregulating the expression of inducible nitric oxide synthase (iNOS) and nitric oxide (NO) production by macrophages [48]. OPN was initially termed as Eta-1 (early T lymphocyte activation gene 1) due to its expression in activated T cells and its crucial role in mediating the induction of immune responses through T cell regulation [49]. In inactivated T cells, *SPP1* expression is regulated by T-bet (a T-box transcription factor necessary for CD4^+^ T helper cell lineage control), which is essential for T_H_1 skewing [49]. OPN is produced in macrophages by pro-inflammatory cytokines, including interleukin 1-beta (IL-1β), interleukin-6 (IL-6), tumor necrosis factor-alpha (TNF-α), and interferon-gamma (IFN-γ), and by another biomolecules such as angiotensin-II, oxidized low density lipoprotein (LDL), and phorbol-ester [50,51]. 

OPN has a critical role in macrophage function, regulating its accumulation and retention at injury sites and facilitating phagocytosis. Moreover, OPN acts as a potent chemoattractant, promoting macrophage migration and stimulating interleukin-12 (IL-12) production, whereas it inhibits apoptosis and interleukin-10 (IL-10) production by interactions with α_4_ and α_9_ integrins and CD44 [14,21,45,50,52]. In lipopolysaccharide (LPS)-stimulated macrophages, OPN can be upregulated by the *SPP1* promotor [47]. In the absence of OPN, macrophages present a diffuse distribution of CD44, leading to less cytokine production and migration [45]. Through CD44 and α_v_ integrin binding, OPN induces maturation and migration of DC, acting as a pro-survival molecule, whereas OPN blockade increases apoptosis and reduces major histocompatibility complex (MHC) class II expression. When OPN activates DCs, it increases the expression of MHC class II, CD80/86, and intracellular adhesion molecule-1 (ICAM-1), which enhance their T_H_1-polarizing ability [14,45]. Due to OPN exerting a T_H_1 cytokine function, it is considered a multifunctional protein that participates in the development of inflammatory diseases.

### 1.4. Osteopontin Involvement in Inflammatory Diseases

OPN is expressed in high concentrations in inflammatory and autoimmune diseases such as Crohn’s disease, cirrhosis, obesity, atherosclerosis, cancer, systemic lupus erythematosus (SLE), multiple sclerosis (MS), atherosclerosis, RA, and OA, among others [37,53].

In Crohn’s disease, OPN is upregulated in intestinal mucosa and detected in higher concentrations in plasma, which is associated with the inflammatory severity regulating the T_H_1 immune response [54,55,56]. In cirrhosis, OPN expression is promoted by the Notch signaling pathway in hepatocytes and mediates liver fibrosis [57]. Moreover, OPN is secreted by hepatic lipid-associated macrophages implicated in non-alcoholic fatty liver disease related to obesity [58]. On the other hand, it has been observed that OPN is expressed in vascular smooth muscle cells in different stages of atherosclerosis, and angiotensin-II induces its expression [59,60].

In malignant processes, OPN and its isoforms are involved in the modulation of tumor-associated inflammation, invasiveness, drug resistance, and poor prognosis; even in specific cancer types, OPN is considered a biomarker [61,62,63]. In autoimmune diseases, OPN exerts its harmful activity by promoting the secretion of IFN-γ and interleukin 17 (IL-17) in T cells and IL-6 in monocytes and supporting T follicular helper (TF_H_) differentiation [64]. In SLE, OPN enhances the inflammatory process-activating T cells, NK, and macrophages driving T_H_1 cell differentiation, and it could be involved in the propagation and differentiation of B cells and autoantibodies production [53]. Regarding MS, OPN promotes the activation and survival of autoreactive T cells and is associated with relapses [65]. In addition, there is evidence for the implication of OPN in chronic inflammatory diseases involving bone and cartilage degradation, such as RA and OA. In the following sections, we describe the mechanism of damage proposed for OPN in these diseases and its regulation by micro RNAs (miRNA) and discuss suggested therapies. 

## 2. A Brief Look at the Basic Immunopathology of Rheumatoid Arthritis and Osteoarthritis

RA is a complex systemic chronic progressive inflammatory disease characterized by inflammation in the synovial membrane and hyperplasia that leads to cartilage and bone destruction [66]. It is a multifactorial disease related to multiple environmental and genetic factors [67]. Clinically, the manifestations of RA in symmetrical joints include arthralgia, swelling, redness, and the involvement of extra-articular organs, such as the heart, kidney, nervous system, skin, lung, eye, and skin, among others [67,68]. 

Joint destruction is the central characteristic of RA, where the articular cartilage is considered the main target of the immunopathological processes [69]. Currently, RA pathogenesis has not been fully elucidated, however, it is known that structural damage, infiltration of immune cells, and the production of proinflammatory cytokines occur in the articular microenvironment [70]. T cells, B cells, and macrophages infiltrate the synovial membrane, eliciting a response in fibroblast-like synoviocytes (FLS) that hyper-proliferate and contribute significantly to inflammation, causing damage to the joint’s architecture [71]. 

On the other hand, OA is a common form of arthritis affecting the elderly, which is characterized by mechanical abnormalities that include degradation of subchondral bone and articular cartilage and osteophyte formation due to the remodeling of the adjacent bone. OA patients clinically show joint stiffness, chronic joint pain, and movement limitation accompanied by inflammation generated by excessive cytokine secretion [72,73]. OA is characterized by the gradual loss of articular cartilage, composed of chondrocytes and ECM. The chondrocyte, which maintains a delicate equilibrium between ECM synthesis and degradation, is mainly composed of aggrecan and collagen type II (COL2), considered essential components of cartilage proteoglycans [74]. In OA, the apoptosis of chondrocytes (induced by stress, activation of death receptors, accumulation of oxygen species, or mitochondrial dysfunction), chondroptosis, necrosis, or a combination has been related to the pathogenesis [75,76,77]. 

As the most common degenerative joint diseases, RA and OA express effector molecules such as cytokines, MPP, and chemokines involved in cartilage and bone degeneration. Table 1 summarizes representative biomolecules and their pathological function identified in in vivo and in vitro RA and OA analyses.

During the RA and OA immunopathologic processes, the synovial tissue is infiltrated by immune cells and cytokines, which leads to an upregulation of integrin receptors and their ligands that enhance the production of MMP and cytokines, cell extravasation, and activation of FLS. This process triggers invasion and degradation of cartilage that thereby generates ECM debris that may further activate integrins [113]. Thus, integrins and their ligands such as ECM proteins contribute to the maintenance of the synovial lining in the immunopathogenesis of RA and OA [114,115]. For instance, in RA, the expression of α_v_β_3_ integrin in synovial tissues facilitates the invasion and attachment of FLS to the cartilage–pannus junction, inducing MMP and cathepsin secretion [116]. Synovial fluid (SF) of OA patients showed an increased expression of α_v_β_3_ that may result in the stimulation of inflammatory mediators and MMP that lead to joint destruction. 

As mentioned in this review, OPN is a member of the RGD-motif proteins that act as a ligand for integrins such as α_v_β_3_. Since α_v_β_3_ is overexpressed in AR and OA, one proposed mechanism through which OPN can exert its pathologic role is attaching α_v_β_3_ and promoting the development of hyperplastic, tumor-like invasive synovitis, contributing to the degradation of bone and cartilage [116]. In the following sections, we focus on the pathological role that OPN may play in the RA and OA cartilage and joint microenvironment, which can have a systemic impact on patients. 

## 3. Osteopontin: Could It Be Considered a Common Denominator in Rheumatoid Arthritis and Osteoarthritis Immunopathology?

### 3.1. Soluble Osteopontin Levels in Rheumatoid Arthritis and Osteoarthritis Patients

OPN is an effector molecule associated with inflammation in many chronic inflammatory and autoimmune diseases, so it has been suggested as a potential agent for promoting joint degradation in RA immunopathogenesis [117]. Clinically, high levels of soluble OPN (sOPN) have been detected in the SF plasma, serum, and urine of RA and OA patients and have been related to clinical severity indexes. Table 2 summarizes sOPN levels in plasma, serum, SF, and urine of RA and OA patients compared to controls.

Plasma and serum sOPN levels coincide with arthritis flares, and, interestingly, during arthritis progression, higher sOPN levels are maintained and decrease after treatment with immunosuppressive drugs or biologics [119,126]. Nevertheless, to establish OPN as a possible diagnostic marker, it is necessary to compare it with reference values. Few studies have determined OPN plasma values in a healthy population, concluding that the OPN physiological levels change during human growth. Nourkami-Tutdibi and colleagues determined that during the neonatal/postnatal period, significantly higher OPN levels are detected (2300 ± 552 ng/mL), and they decrease in adulthood (300 ng/mL) [130]. In addition, previous work had determined an OPN range from 31–200 ng/mL [121]; therefore, it is concluded that these values could be taken as a reference. 

Based on these values, only one result in RA patients [120] and two in OA patients showed higher plasma sOPN levels than the reference [122,123]. Discrepancies could be related to the used technique (available commercial ELISA) and sample conditions [128]; however, significant differences between plasma and serum sOPN levels in AR and OA patients vs. controls were detected by the authors of the studies, independent of baseline values, which could be interpreted as an indicator of the inflammatory status [66,121,124,125,126,127]. In this regard, the above-mentioned could be applied to the results obtained from RA patients categorized as non-responders to treatment that presented plasma sOPN higher levels than the responders [119], and RA patients with severe disease activity showed slightly higher plasma sOPN levels than RA patients with moderate disease activity [126].

Furthermore, studies performed in OA patients revealed that plasma and serum sOPN levels were higher vs. controls [121,124,127], and, when determined in SF, reflected the radiographic severity [72,128]. Interestingly, Abdelnaby et al. and Slovacek et al. performed an analysis in patients undergoing a replacement surgery to quantify the plasma sOPN levels, and they determined that circulating sOPN levels showed no significant reduction after operation and do not respond to replacement treatment; these results indicate that plasma sOPN levels may not be used as a biomarker in the follow-up examination [123,124].

On the other hand, Dong and colleagues detected sOPN-FL and sOPN N-half levels in serum and SF in OA patients and found that the serum sOPN-FL levels and sOPN N-half to sOPN FL ratio in SF correlated with OA severity 125]. In addition, Jiang et al. quantified sOPN N-half in SF in knee OA patients classified with radiographic severities. They observed that sOPN N-half may serve as a biomarker for determining severity and could be predictive of prognosis of knee OA [72]. Regarding RA, Hasewaga and colleagues found that sOPN N-half levels in SF were 30-fold higher compared to OA [9], and Shio et al. determined that sOPN N-half levels in urine were higher in RA compared to OA samples [120]. These results revealed that the sOPN N-half levels in SF may represent the local generation of thrombin that is increased in inflammatory rheumatic diseases. The following section explains in detail the molecular mechanism by which OPN mediates damage to bone and cartilage.

### 3.2. Proposed Molecular Mechanism of OPN in Rheumatoid Arthritis and Osteoarthritis

One of the first studies where OPN was associated with RA pathology was carried out by Petrow and colleagues, where they demonstrated that FLS produced OPN at the sites of cartilage invasion and in the synovial lining layer, inducing the attachment of FLS to the cartilage and producing MMP-1 in chondrocytes that contributes to ECM degradation [131]. These observations were in agreement with Suzuki et al.’s findings, which found that OPN was overexpressed in the RA synovial sublining and lining layers, in the cartilage interface, and in invading synovium, in addition to the observations of OPN mRNA overexpressed in CD4^+^ synovial T cells that correlates with the expression of CD44 and α_v_β_1_ integrin receptor [132,133]. 

Considering that RA is a specific T_H_1 disease, OPN may also contribute to the initiation/onset of arthritis by polarizing T_H_1 cytokine responses and bone resorption by OC [117]. Xu and colleagues observed a high mRNA OPN expression in CD4^+^ synovial T cells that correlates with OPN levels in SF. This OPN overexpression pattern was observed to be limited to the rheumatoid synovium. Thus, OPN in FLS is associated with the local inflammatory milieu [134]. Subsequently, in a study by Zheng et al., they noticed that elevated concentrations of sOPN correlate with inflammation marker levels in the serum, such as macrophage inflammatory protein-1 beta (MIP-1β) and monocyte chemo-attractant protein 1 (MCP-1) in monocytes. According to the authors, OPN induces the expression in CD14^+^ monocytes of MIP-1β and MCP-1 through its structural motif located at the 50–83 residues of human OPN. The chemokine expression induced by OPN was mediated through the NF-kB pathway and involved IKKβ, protein 38 (p38) mitogen-activated protein kinases, and c-Jun NH_2_-terminal kinase (JNK) activation [135]. 

In RA, it has been observed that the increased expression of thrombin in SF is considered an essential player in the coagulation cascade and the leading producer of the OPN-N half. Thrombin sustains inflammation; it could originate from the blood and can be present in joint space due to a leakage, although it is recognized that it may also be generated locally [38,136]. OPN N-half has been found in high concentrations in SF and plasma of RA patients, whereas OPN N-half and OPN-FL ratios are significantly increased in RA SF and plasma compared with controls [9,137]. Moreover, in studies by Shio and colleagues, OPN N-half in urine from active RA patients was found in higher concentrations with RA and correlated with inflammation markers such as erythrocyte sedimentation rate, C reactive protein, and rheumatoid factor. In addition, urine sOPN N-half was detected at significantly higher levels in RA patients with progressive bone degradation compared to RA patients in the early stage of bone destruction [120]. 

OPN could play an essential role by promoting T_H_1 and T_H_17 cell differentiation. Specifically, OPN levels are correlated with IL-17 secretion, T_H_17 cell incidences in RA patients’ SF, and inflammation parameters. Furthermore, OPN promotes in osteoblasts the expression of IL-17, enhancing the migration via the Syk/PI3K/Akt signaling pathway of monocytes [138,139]. Moreover, it was observed that the phosphorylated-OPN form (p-OPN) increases the OC and macrophage activation. This process is controlled by extracellular tartrate-resistant acid phosphatase (TRAcP). Luukkonen and colleagues elucidated that when TRAcP 5B is produced in insufficient quantities in synovial tissue, it led to an excessive concentration of p-OPN associated with OC activation, cartilage degradation, and activation of immune cells [140]. In a recent study, Xie et al. noticed that the highest sOPN serum levels were observed in RA patients in an active phase of the disease and correlated negatively with Treg absolute counts [141]. 

Regarding OPN and RA treatment, it has been observed that low sOPN levels at baseline predict clinical remission one year after initiating tocilizumab treatment in a prospective analysis of biologic-naïve RA patients [142]. These findings agree with Sennels and colleagues’ previous findings, where RA patients with the active disease showed an increased circulating level of sOPN that did not decrease during etanercept therapy [143].

The role of OPN in OA pathology was described by Pulling et al. in one of the first assays performed on osteoarthritic cartilage and bone samples. The authors reported that OPN mRNA expression was detected in osteoblasts in subchondral bone and increased with osteoarthritic severity and cartilaginous matrix degradation. Additionally, they observed that the most substantial OPN mRNA detection was found in clusters of proliferating chondrocytes and the deep zone of cartilage [144]. These findings were subsequently confirmed by two different studies conducted by Yagi et al. and Sanchez et al. In the first analysis, it was found that OPN mRNA expression showed a 3.6-fold increase in stage-advanced OA, and, in the second assay, the authors detected that, in sclerotic osteoblasts, OPN gene expression was up-regulated [145,146]. Further, Dai and colleagues suggested that the increase of OPN mRNA in the subchondral bone was associated with OA severity and correlated with the ratios of p-Smad2/3^+^ in OC [95]. 

On the other hand, when OPN in cartilage was found at high levels, it activated the NF-kB pathway related to production of cytokines such as IL-6, IL-1β, and IL-8, and chemokine secretion, such as CC motif chemokine ligand 2 (CCL2) and CXC motif chemokine ligand (CXCL1), which lead to NO and prostaglandin E_2_ (PGE_2_) production. When these cytokines and mediators are produced in high concentrations, they produce an imbalance of cartilage homeostasis, exert harmful effects on chondrocyte functions, and are associated with progressive articular degradation [147].

Most published articles agree that OPN is considered an essential regulator of OA progression and may even be associated with the degree of pain. Jiang et al. determined that during chondrocyte maturation, OPN is essential in the cartilage-to-bone transitions, therefore, it could be involved in cartilage degradation. Kulkarni and colleagues observed in a proteome analysis that OPN showed 35.96-fold higher expression levels in OA SF in Kellgren–Lawrence (KL) grade IV samples [148]. Tanaka et al. observed that OPN decrease in bone is essential for promoting vulnerability to hip fracture [149]. Additionally, Yamaga et al. found that the severity of articular cartilage damage after anterior cruciate ligament (ACL) and joint pain was correlated with OPN levels in SF. They noted that both OPN-FL and OPN N-half levels in SF correlated with pain severity within a month after ACL rupture. 

Since OPN can function as a T_H_1 cytokine and regulates prostaglandin-endoperoxide synthase 2 (PTGS_2_) and iNOS expression, it was speculated that OPN might mediate the expression of molecules related to the induction of pain, such as PGE_2_ and NO, after ACL rupture during the acute inflammatory stage [150]. Regarding the higher level of phosphorylation of OPN, Xu and colleagues observed that p-OPN led to elevated expression of MMP-13, unlike an unphosphorylated OPN [151]. Thus, the role of OPN in cartilage and bone destruction has been demonstrated in patients, where both protein and mRNA expression levels were detected in the highest concentrations compared to healthy controls. However, studies are required to affirm that OPN can be considered as a biomarker of disease progression and prognosis, and even an indicator of response to treatment. 

### 3.3. SPP1 Genetic Variations Associated with Rheumatoid Arthritis and Osteoarthritis

Genetic variations in *SPP1* are identified in the 5′ untranslated region (UTR), introns, exons, and 3′UTR sites and have been associated with individual susceptibility, development, and chronic inflammatory disease activity such as cancer [64]. In RA, discrepancies exist because some analyses did not find a correlation between *SPP1* single nucleotide polymorphism (SNP) and sOPN levels or individual susceptibility [152,153]. Conversely, there are investigations that associate *SPP1* SNP with RA susceptibility, joint destruction, the course of oligoarticular onset juvenile idiopathic arthritis (JIA), and the positivity to anti-citrullinated protein antibodies (ACPA) [154,155,156,157]. Concerning OA, most studies associate *SPP1* genotype carriage with OA development risk and radiographic severity [72,158,159,160]. Regarding *SPP1* variations, it is essential to note that differences are detected between populations in the same, for instance, rs11730582; this may be related to environmental factors, lifestyles, and genetic backgrounds that determine susceptibility to diseases. Studies in different populations and larger sample sizes are needed to determine whether OPN genetic variations in humans are associated with the susceptibility to the development of rheumatic diseases. Table 3 summarizes the *SPP1* SNP analysis performed in AR and OA patients. 

### 3.4. In Vitro Studies Reveal the Crucial Role of Osteopontin in Rheumatoid Arthritis and Osteoarthritis Pathology

One of the first in vitro assays to determine OPN levels in cultured synovial cells from RA and OA patients was performed by Ohshima and colleagues, where they observed that OPN showed a marked increase in SF of RA and OA compared to controls [117]. Since OPN is a protein susceptible to multiple modifications, in vitro analysis is crucial for elucidating the function of OPN in RA and OA immunopathology. Several in vitro studies focused on the interaction between FLS and leukocytes, studying the findings in cocultures. When the FLS obtained from RA synovium is cocultured with B lymphocytes (FLS-B), the adhesion of B cells to FLS supports the B lymphocytes’ survival and enhances the production of cytokines and immunoglobulins. In an FLS obtained from RA patients, Take et al. detected a specifically modified 75 kilodalton (kD) form of OPN with significantly higher IL-6 production in FLS-B lymphocyte cocultures. In addition, they found that the 75 kD OPN formed a >200 kD OPN/fibronectin-crosslinked molecule via transglutamination that supported adhesion of B lymphocytes to FLS in FLS-B lymphocyte cocultures, enhanced IL-6 production, and was associated with the cell surface of the plasma membrane [161]. Further assays performed by Mehta and colleagues focused on the OPN–fibronectin interaction. They showed that this interaction is relevant for the expression of inflammatory mediators by B lymphocytes that infiltrate the synovial joint because the FLS are capable of expressing surface-bound fibronectin, which forms covalent crosslinks with OPN N-half and promotes the infiltration of B cells through engagement of VLA4 (very late antigen-4) expressed on B cells surfaces [162]. Thus, OPN can interact with other proteins to exert proinflammatory functions, such as fibronectin and tenascin-C (TN-C). Asano and colleagues demonstrated that OPN and TN-C contribute to RA by interacting with α_9_β_1_ integrin in FLS and macrophages isolated from synovial tissue [163]. 

In contrast, it has been recognized that the OC plays a significant role in the pathogenic bone destruction in RA due to bone-resorbing giant polykaryon cells that fuse to form multinucleated giant cells (MGCs). MGCs and immature dendritic cells (iDCs) form OC-like MGCs involved in the osteolytic lesions observed in inflammatory bone diseases such as RA to mediate bone resorption. It has been observed that OPN plays a crucial role in modulating OC function through the binding with α_9_β_1_ integrin and during OC-like MGC formation from iDCs, because a large amount of OPN (both mRNA and protein) was detected in cultured cells. Moreover, OPN-FL stimulates cells that have differentiated into OC-like MGCs, while OPN N-half stimulates cells that have retained the character of iDCs [164]. Subsequently, in a chemotaxis assay performed by Shi and colleagues that measured in the presence of OPN during the migration of leukocytes, they noted that PI3K, JNK, and ERK were involved in OPN expression induced by LPS. Moreover, they observed that OPN could induce MCP-1 overexpression through OPN-integrin β1-JNK/p38 pathway activation [165]. In addition, it has been observed that high extracellular calcium concentrations in the SF of RA are related to activated cells and, in other cases, cell death. The extracellular calcium danger signals activate the NLR family pyrin domain-containing 3 (NLRP3) inflammasome through the calcium-sensing receptor (CaSR) in monocytes/macrophages mediated by calciprotein particle (CPP) uptake. The CPPs and the extracellular calcium can differentiate the monocytes of the synovia into calcium macrophages that produce, in a constitutive manner, OPN mRNA and protein and secrete M1 pro-inflammatory molecules [166]. The constitutive expression of OPN is crucial in favoring its pathological functions because it is considered an inducible molecule; however, in certain diseases, its transcription and translation are significantly upregulated [167]. 

Concerning the in vitro assays to discern the implication of OPN in OA immunopathology, various experiments have focused on analyzing FLS- and OA-associated cytokines, such as hyaluronic acid (HA), OPN, and its receptor CD44. The HA forms the core component of the ECM, and changes in its molecular mass and concentration are found in osteoarthritic joints and SF, contributing to the low-level inflammatory condition that characterizes OA. OPN and HA can bind to the CD44 receptor, significantly increasing synovial tissue in OA patients. Therefore, changes in HA-binding protein CD44 levels might be involved in the immunopathology of OA. Zhang and colleagues demonstrated in FLS derived from the knees of OA patients that HA can upregulate OPN mRNA, promoting synovitis. However, they noted that HA could not affect the CD44 mRNA expression. Thereby, synovitis promotes OPN expression that can increment the expression of CD44 and HA [168]. Moreover, when Yuan et al. investigated the CD44 effect on HA expression in cultured human knee OA chondrocytes, they found that OPN can promote the expression of hyaluronan synthase 1 (HAS1) and induce HA expression by reacting with the CD44 ligand receptor [169]. 

Furthermore, they observed that OPN increased α_v_β_3_, HA, and HAS expression, proposing that the OPN/α_v_β_3_/HAS1/HA axis is essential in the regulation of HA expression in chondrocytes [170]. Following the above, Zhang and colleagues found in cartilage cell culture that the expression of OPN is increased and is capable of upregulating the MMP-13 expression level and downregulating the production of COL2 and cartilage oligomeric matrix protein (COMP) [171]. Regarding OPN and angiogenesis, Xu et al. demonstrated in cultured chondrocytes that OPN induces the expression of vascular endothelial growth factor (VEGF) through PI3K/AKT and ERK1/2 phosphorylation [172]. Katayama and colleagues determined that OPN-specific phosphorylation is critical for OC interaction, compared to with osteoblasts [173]. In addition, it was observed that p-OPN mediates apoptosis and the secretion of inflammatory molecules such as IL-6, IL-1β, TNF-α, and NF-kB in human knee OA chondrocytes. Thus, OPN in a phosphorylated state promotes a proapoptotic and proinflammatory process that contributes to OA immunopathology [76].

Nonetheless, it also was determined that OPN could not be involved in the progression of OA and may even have a protective role. One study by Cheng et al. noted that OPN inhibits hypoxia-inducible factor 2 α (HIF-2α) mRNA expression in chondrocytes from knee OA patients. Thus, OPN confers cytoprotection against hypoxia/reoxygenation-induced apoptosis through CD44 interaction [75]. In addition, Gao and colleagues observed that OPN activation in chondrocytes suppresses A disintegrin and metalloproteinase with thrombospondin motifs-4 (ADAMTS4) expression and avoids aggrecan degradation, reducing cartilage degeneration [74]. Tian et al. found that deficiency of OPN in OA chondrocytes enhanced the apoptosis, senescence, IL-1β, TNF-α, MMP-13, collagen 10 alpha 1 (COL10A1), and ADAMTS5 expression, but decreased collagen 2 alpha 1 (COL2A1) expression [77], and, recently, Liu and colleagues demonstrated that OPN delays chondrocyte degeneration and reduces cartilage matrix by acting as an activator for PI3K signaling through its binding to CD44 [174]. 

Thus, in vitro studies revealed the participation of OPN in synovial cells and chondrocytes, and showed that, in addition to forming complexes with proteins, OPN stimulates the differentiation of immune cells and specific receptors related to the induction of inflammation associated with cartilage and subchondral bone degradation. Nevertheless, some studies have not ruled out the possibility that OPN may be protective against cartilage degeneration; further evidence is needed to prove it. 

### 3.5. Role of Osteopontin in Experimental Models of Rheumatoid Arthritis and Osteoarthritis

The experimental models have been considered an essential part of the research to elucidate the OPN role in human pathology. One of the first experimental tests of the participation of OPN in the regulation of bone metabolism was provided by Yoshitake and colleagues, who reported that OPN knockout (KO) mice were resistant to bone-related-to-estrogen depletion after ovariectomy in comparison to wild-type (WT) mice and showed a 10% reduction in the volume of trabecular bone after ovariectomy compared to a 60% reduction noted in WT [175]. Subsequently, several researchers have conducted studies based on the collagen-induced arthritis (CIA) model, which is a suitable model in which there is the possibility of analyzing the arthritis progression induced by injection with a mixture of anti-COL2 monoclonal antibodies followed by LPS treatment. 

One of the first studies carried out to evaluate the role of OPN in CIA was conducted by Yumoto and colleagues, in which they showed that in the absence of OPN, the apoptosis in articular chondrocytes was suppressed and the joint swelling was suppressed and was associated with the CD31^+^ vessels reduction in synovia. These results are essential for understanding the involvement of OPN in RA angiogenesis [69]. 

A subsequent study by Oshima et al. determined that the OPN mRNA and protein were present at bone erosion sites where activated OC was detected. In addition, they detected that the α_v_β_3_ integrin was found coincident with OPN sites of the bone–pannus junction [137]. Yamamoto and colleagues found that the cryptic epitope SLAYGLR, exposed by OPN N-half, is involved in the pathogenesis of the CIA model because monocytes expressing α_9_ and α_4_ integrins recognize the sequence and exhibit a significant capacity for cell migration [176]. Kanayama and colleagues found that OPN N-half, TN-C, and α_9_ integrin are expressed for FLS in arthritic joints. The α_9_ integrin is related to the production of cytokines and chemokines, the induction of the trans-endothelial migration of neutrophils, and the generation of OC. [177]. Furthermore, they identified that α_9_ integrin signaling induced by OPN N-half and TN-C promoted the production of T_H_17-related cytokines by conventional DC cells and macrophages in synergy with Toll-like receptor (TLR)-1 and -4 signaling, leading to T_H_17 cell differentiation and arthritis development [178]. Figure 2 represents the OPN role in RA immunopathology based on clinical, in vitro, and experimental observations.

On the other side, Su et al. determined that in primary osteoblasts, OPN induces up-regulation of oncostatin M, an IL-6 family member capable of stimulating bone formation associated with the modulation of RA and OA. Interestingly, they found that lentiviral knockdown of OPN significantly abrogated inflammation and bone erosion in the CIA’s model [70]. Most of the results have pointed out that OPN is one of the CIA model’s principal mediators of bone erosion. However, it has been observed that OPN is not required to induce bone erosion, cartilage damage, and inflammation in the autoantibody-induced arthritis model. Jacobs and colleagues determined the OPN function in the K/BxN serum-transfer arthritis model that bears marked clinical and histologic similarity to RA. Although, during K/BxN serum-transferred arthritis, OPN was observed to be up-regulated, the authors conclude that OPN is not an inducer of bone erosion in antibody-mediated arthritis [179]. Moreover, the implication of OPN in OA immunopathology has also been assessed in experimental models. Matsui et al., in OPN KO instability-induced and aging-associated OA models, noticed the induction of MMP-13 that degrades COL2 and an increased proteoglycan loss from cartilage tissue [180]. Furthermore, Almonte and colleagues identified in a rat experimental OA model that in the middle and deep zones of cartilage, α_4_ integrin expression was increased and correlated with the abnormal endochondral ossification of the cartilage through OPN interaction [181]. In addition, Gao et al. determined in an OA rabbit model of the posterior cruciate ligament that OPN N-half was detected in high concentrations in SF and correlated with macroscopic and histological OA scores that determine cartilaginous damage degree [182]. On the other hand, Martínez-Calleja and colleagues observed in a rat model of OA calcium deposits and overexpression of OPN and calcium deposits in the cartilage of osteoarthritic rats, specifically in the superficial zone of articular cartilage in the early stages [183]. 

In an OA model established in rats, Sun et al. determined that OPN was upregulated and promoted the proliferation of primary chondrocytes, whereas the rats treated with OPN knockdown adenovirus showed alleviated cartilage [184]. Moreover, in a recent study, Lin and colleagues found in a destabilization OA mouse model that OPN overexpression is related to h-type vessel formation in the subchondral bone [185].

Figure 3 represents the involvement of OPN in OA immunopathology based on clinical, in vitro, and experimental observations.

From the data obtained from both RA and OA patients, in vitro analyses, and experimental models, we concluded that OPN could be considered a common denominator molecule in the immunopathogenesis of inflammatory rheumatic diseases since it is a crucial mediator of intercellular communication and an influential factor in the microenvironment regulation of inflamed joints. 

## 4. Osteopontin Regulation by MicroRNA in Rheumatoid Arthritis and Osteoarthritis 

microRNA (miRNA) is defined as a small single-stranded non-coding RNA molecule of around 22 nucleotides in length that negatively modulates gene expression, inhibiting translation and degrading mRNA and positively increasing the translation rate. Some attributable functions of miRNAs are their participation in various processes such as immune response, proliferation, metabolism, homeostasis, apoptosis, and senescence, among others [186]. Regarding OPN regulation by miRNAs in RA, it was observed that OPN is directly targeted by miR-539, which acts as an inhibitor of OPN mRNA translation. Liu et al. analyzed, in a dual-luciferase assay in immortalized FLS, that immortalized MH7A cell line transfected with agomiR-539 and OPN knockdown by small interfering RNA (siRNA) miR-539 regulates OPN expression through 3′UTR complementary binding [187]. Studies performed by Tsai and colleagues in osteoblasts derived from RA patients revealed that OPN negatively regulated miR-129-3p expression via the Syk/P13K/Akt signal cascade. These findings were confirmed in CIA mice using lentiviral vectors that inhibited OPN expression and ameliorated the ankle’s joint cartilage degradation, articular swelling, and monocyte infiltration [139]. On the other hand, Qian et al. discovered that by silencing OPN using lentiviral-OPN short hairpin RNA, the tartrate-resistant acid phosphatase-positive OC and the bone erosion extent in the adiponectin-treated CIA mice were reduced [188]. 

In OA, it has been observed that miR-127-5p contributes to OA development and is a critical MMP-13 regulator in human chondrocytes. Tu and colleagues noticed that miR-127-5p in OA patients is downregulated and downregulates OPN expression through binding OPN mRNA at the 3′UTR site. Thus, they conclude that miR-127-5p could represent an alternative to control the establishment and development of OA [189]. Furthermore, Lin et al. demonstrated in OA chondrocytes that miR-186 negatively targets *SPP1*, a regulator of the PI3K-AKT pathway. The authors suggest that miR-186 affected chondrocytes by targeting and depleting *SPP1*, inhibiting chondrocyte apoptosis and regulating the PI3K-AKT pathway in mice with OA. Thus, restoring miR-186 might be a future therapeutic strategy for OA [190]. 

Current research has been interested in a class of long non-coding RNAs (lncRNAs) that have shown that their altered expression can lead to gene disorders that contribute to the regulation of biological functions and the development of diseases. The lncRNAs are RNAs of 200 nucleotides in length that lack an open reading frame, are classified as a non-protein-producing RNA, and are named following their host protein-coding gene. In OA, it has been determined that lnRNAs exert essential roles in cartilage and bone development, promoting ECM cartilage degradation when detected in abnormal expression [191]. Studies conducted by Wang and colleagues observed that miR-181c inhibition promoted OA progression and increased OPN protein levels. This process was demonstrated by observing that miR-181c ectopic expression repressed synoviocyte proliferation and IL-6, IL-8, MMP-13, and OPN levels. Additionally, the authors searched for an lnRNA candidate to rescue miR-181c expression and found that nuclear-enriched abundant transcript 1 (NEAT1) regulated miR-181c inversely and competed with OPN mRNA for its binding. They noticed that after NEAT1 knockdown, the synoviocyte proliferation and OPN protein levels were repressed, and the MMP-13, IL-6, and IL-8 levels were reduced. Therefore, the author considered that targeting NEAT1 might be an efficient OA treatment strategy [192]. Liang et al. observed that miR127-5p binds to the 3′UTR of OPN, inhibiting OPN expression and chondrocyte proliferation. However, the miR-127-5p expression is downregulated in OA tissues. The lncRNA metastasis-associated lung adenocarcinoma transcript 1 (MALAT1) knockdown repressed human OA chondrocyte proliferation and OPN significantly [193]. 

In contrast, it has been noted that in OA tissue cartilage samples, lncRNA X-inactive-specific transcript (XIST) and OPN mRNA were upregulated. XIST competes with OPN for miR-376c-5p binding, and the inhibition of miR-376c-5p reverses the effect of XIST silence. Hence, it was suggested that the XIST/miR-376c-5p/OPN axis can regulate the OA inflammatory microenvironment [194]. Moreover, it was observed in primary chondrocytes that OPN induces the lncRNA homeobox transcript antisense RNA (HOTAIR) expression associated with chondrocyte proliferation [195]. In addition, it was remarked that miR-181a-5p overexpression targeted OPN and can inhibit OA chondrocyte viability. Zheng and colleagues observed that through binding miR-181a-5p, lncRNA myocardial infarction-associated transcript (MIAT) negatively regulated miR-181a-5p expression and partially attenuated miR-181a-5p inhibition upon the expression level of OPN, suggesting that MIAT competitively binds to miR-181a-5p to counteract miR-181a-5p-mediated suppression of OPN. These findings conclude that the lncRNA MIAT/miR-181a-5p/OPN axis could regulate chondrocyte proliferation and apoptosis [196]. 

Thus, altered expression of miRNAs and lncRNAs might play functional roles in the disorders of OPN expression and synoviocyte proliferation, leading to RA and OA progression.

### Are They Blocking Osteopontin Activity: An Alternative for Rheumatoid Arthritis and Osteoarthritis Treatment?

Because OPN has shown a critical role in RA and OA immunopathogenesis, several trials have been performed to determine if the suppression of OPN activity could be a potential therapeutic target suitable for rheumatic patients. One of the first experimental assays was conducted by Yamamoto and colleagues, where, in a CIA model, they employed the rabbit anti-mouse antibody M5 that reacted to the SLAYGLR murine sequence and observed that M5 in arthritic joints abrogate bone degradation, monocyte migration towards the OPN N-half inhibiting synovium proliferation, and inflammatory cell infiltration [176]. Subsequently, the authors designed and generated a murine monoclonal antibody, termed C2K1, that recognized the SVVYGLR epitope in humans and observed that C2K1 could ameliorate the established CIA in cynomolgus monkeys in therapeutic administration [197]. Furthermore, Fan et al. noted that the monoclonal antibody 23C3 against human OPN effectively inhibited CIA development and reversed established disease damage in DBA/1J mice [198]. 

Analysis performed by Boumans and colleagues using ASK8007, a humanized monoclonal immunoglobulin G (IgG1) antibody directed against SVVYGLR epitope in humans, revealed that it effectively inhibited both RGD and α_9_β_1_ integrin-dependent cell binding and showed favorable toxicology results in a monkey CIA model. Nevertheless, when ASK8007 was administrated in RA patients with active, moderate, and severe disease, it did not affect the inflammation or joint destruction; however, it increased the OPN shedding from the endothelium by binding to OPN and forming OPN–ASK8007 complexes, resulting in systemic and partly bioactive OPN accumulation. Therefore, it was determined that blocking OPN was tolerated but did not induce a clinical improvement in RA patients [199]. 

Concerning the negative results described above, Farrokhi et al. demonstrated through preliminary pharmacokinetic/pharmacodynamic modeling and simulation that high-dose intravenous infusion with very short dosing intervals is required to inhibit OPN activity effectively. These results may help explain why ASK8007 failed to exert efficacy in RA patients in clinical trials [200]. On the other side, Mehta and colleagues conducted experiments intending to block the OPN–fibronectin, selecting a single chain variable fragment antibody (scFv). The scFv 31 was employed in FLS derived from RA patients and appeared to significantly reduce the FLS migration, fibronectin production, cell morphology alteration, and actin stress fiber arrangement. In the CIA model, scFv 31 appeared to prevent arthritic damage by inhibiting loss of articular cartilage and synovial hypertrophy, decreasing fibronectin polymerization, and expressing proinflammatory cytokines related to RA [162]. Takanashi et al. synthesized a cream formulation termed GeneCream with an effective topical delivery system containing siRNA that targets OPN. In CIA-treated mice, the GeneCream was applied to the skin, was influential in the delivery of active OPN siRNA, and prevented irreversible bone and cartilage damage [201].

In the experimental model of OA, to date, two studies considered employing natural compounds to achieve a therapeutic effect based on OPN regulation. Bai and colleagues demonstrated that artesunate (a derivate of artemisin from *Artemisia apiacea*) decreased sOPN levels in serum and SF in an OA rat model [73]. Furthermore, Tsai et al. noted that isorhamnetin (herbal flavonoid produced by medicinal plants such as *Ginkgo biloba* L. and *Hippophae rhamnoides* L.) significantly reduced the sOPN levels in monosodium iodoacetate (MIA)-induced OA rats [202]. 

In summary, there are a variety of alternatives for regulating OPN function in both RA and OA; nevertheless, the long-term side effects are still unknown. Further studies are needed to prove the efficacy and safety of antibodies and, in addition, are required to demonstrate that blocking OPN function will not have consequences on the maintenance of the organism’s homeostasis.

## 5. Concluding Remarks

Based on the above background, we conclude that the evidence to consider OPN as a common denominator of RA and OA is forceful, as it contributes to bone/cartilage deterioration and synovitis through mechanisms such as induction of proinflammatory cytokines, MMP production, and induction of apoptosis and angiogenesis. Specific mechanisms have been elucidated in RA, these being induction of T cell differentiation towards a T_H_17 profile, promoting B cell adhesion to FLS to produce IL-6, and induction of monocyte chemokine expression. In contrast, in OA, OPN has been associated with an imbalance of cartilage homeostasis, HA production, COL2 dysregulation, accelerated subchondral bone turnover and remodeling, and even joint pain. 

Due to the crucial role of OPN in RA and OA, too many efforts have been made to determine the mechanisms that can regulate OPN function, such as employing lncRNAs and antibodies to block its activity. In some cases, the results obtained in experimental models have been promising concerning the evident improvement in bone/cartilage deterioration and the downregulation of proinflammatory parameters. However, some studies suggest that OPN may exert a protective function in OA. Therefore, we consider that further evidence is required to determine a therapeutic strategy for regulating OPN function in humans regarding the long-term side effects. 

## Figures and Tables

**Figure 1 biomolecules-13-00502-f001:**
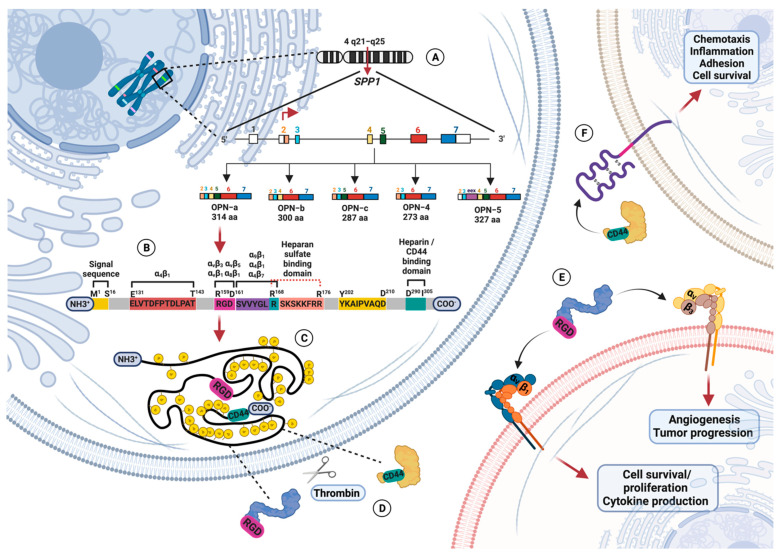
Representation of osteopontin gene, protein structure, and receptor binding. (**A**) Human OPN-encoding gene (*SPP1*) is located on the chromosome 4q21-25 and gives five messenger RNA, where OPN-a is the version with coding information from all exons. (**B**) OPN full-length (OPN-FL) is a 314 amino acid glycosylated phosphoprotein that contains the Arg-Gly-Asp (RGD) motif sequence, SVVYGLR domain, thrombin cleavage sequence, heparan sulfate binding domain, and heparin/CD44 binding domain. (**C**) OPN is heavily posttranslationally modified by serine/threonine phosphorylation, *O*-glycosylation, and tyrosine sulfation. (**D**) Thrombin cleavage changes the OPN conforming N-terminal fragment (OPN N-half), which allows access to the RGD motif. (**E**) OPN N-half interacts with integrins α_v_β_1_ and α_v_β_3_ to exert physiological and pathological functions. (**F**) OPN interaction with CD44 occurs through its putative binding domain located in the last 18 amino acids at the highly conserved C terminus. Created by Biorender with the CT24XI5TS7 agreement number.

**Figure 2 biomolecules-13-00502-f002:**
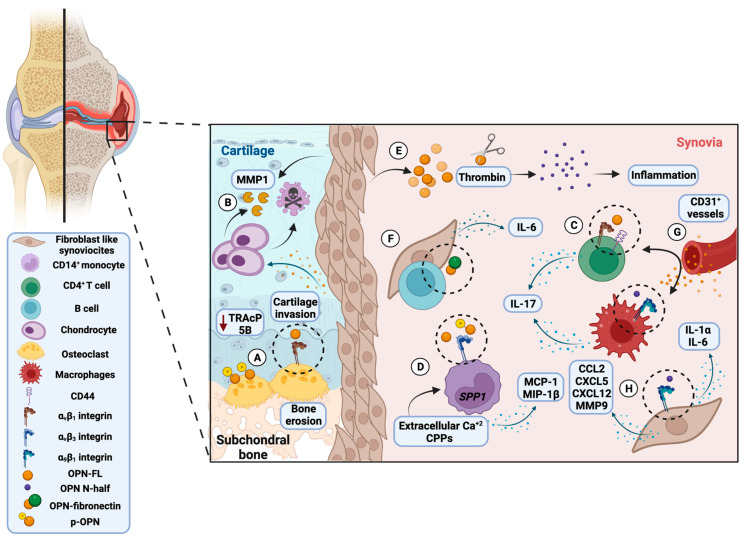
Osteopontin’s role in the rheumatoid arthritis immunopathology, based on clinical, in vitro, and experimental observations. (**A**) Osteoclast overexpresses α_v_β_3_ integrin that binds OPN and promotes bone erosion. Phosphorylated OPN (p-OPN) activates osteoclast due to insufficient tartrate-resistant acid phosphatase (TRAcP) 5B production. (**B**) Fibroblast-like synoviocytes (FSL) produce OPN in the synovial lining layer and at the sites of cartilage invasion, contribute to matrix degradation by stimulation of metalloproteinase-1 (MMP-1) secretion by chondrocytes, and drive chondrocyte apoptosis. (**C**) CD4^+^ T synovial cells express OPN that, through α_v_β_1_ integrin and CD44 binding, induce T cell differentiation towards TH1 and TH17, and secrete IL-17. Furthermore, through α_9_β_1_ integrin binding with OPN N-half, synovial macrophages promote IL-17 production. (**D**) In CD14+ monocytes, OPN induces the expression of monocyte chemoattractant protein-1 (MCP-1) and macrophage inflammatory protein 1 beta (MIP-1β), while p-OPN induces its activation. Extracellular Ca^+2^ and calciprotein particles (CPPs) differentiate monocytes into *SPP1*/OPN calcium macrophages. (**E**) Higher thrombin levels and OPN N-terminal fragment (OPN N-half) were detected in synovial fluid. (**F**) OPN/fibronectin supports the adhesion of B lymphocytes to FLS and enhances IL-6 production. (**G**) OPN promotes angiogenesis through new CD31^+^ vessel formation. (**H**) OPN N-half binds α_9_β_1_ integrin expressed on FLS and promotes the T_H_17-related cytokines by conventional dendritic cells (DC) and macrophages. Created by Biorender with the MN24XI1DEE agreement number.

**Figure 3 biomolecules-13-00502-f003:**
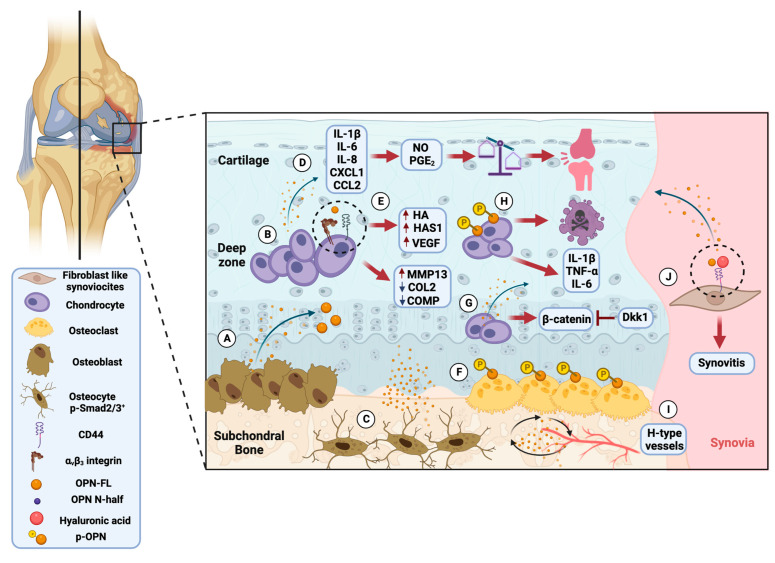
Osteopontin’s role in the osteoarthritis immunopathology based on clinical, in vitro, and experimental observations. (**A**) Osteoblast expresses OPN in the subchondral bone. (**B**) OPN expression is found in deep zone chondrocytes. (**C**) OPN expression correlates with p-Smad2/3^+^ osteocytes associated with OA severity. (**D**) OPN in cartilage produces cytokines and chemokines that lead to the production of nitric oxide (NO) and prostaglandin E_2_ (PGE_2_), which leads to an imbalance in cartilage homeostasis and is associated with joint pain. (**E**) OPN promotes the expression of hyaluronan synthase 1 (HAS1) and hyaluronic acid (HA) in chondrocytes through CD44 and αvβ3. In addition, OPN can upregulate metalloproteinase-13 (MMP-13), downregulate collagen type 2 (COL2) and cartilage oligomeric matrix protein (COMP), and enhance the production of vascular endothelial growth factor (VEGF). (**F**) Phosphorylated OPN (p-OPN) is important for interaction with the osteoclast. (**G**) OPN and β-catenin are increased and could be inhibited by Dickkopf-related protein 1 (Dkk1) in chondrocytes. (**H**) p-OPN causes apoptosis and production of inflammatory cytokines in chondrocytes. (**I**) OPN accelerates the subchondral bone turnover and remodeling and induces the formation of h-type vessels. (**J**) In fibroblast-like synoviocytes (FLS), HA can upregulate OPN and promote synovitis. Created by Biorender with the FQ24XI1ZZF agreement number.

**Table 1 biomolecules-13-00502-t001:** Representative biomolecules involved in rheumatoid arthritis and osteoarthritis immunopathology.

	Ref.	Disease	Immunopathogenic Function
Cytokine			
			Immunopathogenesis key regulator. Activates
	[78]	RA	endothelial cells and recruits proinflammatory
TNF-α			cytokines such as IL-6 and IL-1β.
	[79]	OA	Leads joint and cartilage degeneration by
			collagen disruption and proteoglycan degradation.
			Crucial role in inflammation process in joints, in
	[80]	RA	OC-mediated bone resorption and pannus
IL-6			development.
	[81]	OA	Suppresses proteoglycan production, induces
			oxidative stress, and increases ROS production.
			Mediates bone erosion through induction
	[82,83]	RA	of osteoclastogenic Treg, Promotes the upregulation
IL-1β			of TNF-α and IL-17.
			Drives MMP-9, ADAMTS-4, ADAMTS-5,
	[84,85,86]	OA	and RANKL production associated with
			cartilage catabolism and matrix degradation.
			Promotes TNF-α and IL-17 production and osteoclastogenesis.
	[87,88]	RA	Stimulates and activates T and NK cells, neutrophils, and
IL-15			macrophages.
	[79,89]	OA	Enhances MMP-9 production and induces MMP-1
			and MMP-3 secretion from cartilage chondrocytes in vitro.
			Related to an OC and osteoblast imbalance,
	[90]	RA	it promotes MMPs release in synoviocytes, causing
IL-17			joint destruction.
	[91]	OA	Induces the release of chemokines by chondrocytes
			and FLS, contributing to cartilage collapse.
			Stimulates OC formation by upregulating RANKL
	[78]	RA	production from T cells in RA synovitis. Role in
IL-18			onset and maintenance of inflammatory response.
	[79]	OA	Induces the synthesis of MMP-1, MMP-3, and MMP-13,
			enhancing cartilage degradation.
			Activates T and B cells, monocytes/macrophages, and
	[78,92]	RA	FLS through JAK/STAT, MAPK, and PI3K/Akt signaling
IL-21			pathways.
	[93]	OA	Produced by T_FH_ cells, contributes to inflammation and
			progression of OA stages.
			Promotes T_H_17 differentiation and regulates
	[94]	RA	Treg function. TGF-β1 induces IL-6 production
TGF-β			in arthritic synovium.
			Alters the osteogenic activity of MSC. Generates
	[95]	OA	early non-coupled osteogenesis/osteoclastogenesis,
			accelerating local bone island formation.
MMP			
MMP-1/MMP-3	[96,97]	AR	Peripheral levels can predict progression
			of joint destruction.
			Promotes vascular invasion, degrades cartilage matrix,
MMP-3	[98]	OA	induces OC differentiation, and facilitates inflammatory
			cell accumulation in articular cartilage.
			Contributes to FLS survival, proliferation, migration, and
	[99,100]	AR	Invasion, promoting joint destruction.
			Stimulates TNF-α, IL-6, and IL-8 production, and its
MMP-9			SF levels can predict radiographic progression.
	[101]	OA	Downregulates COL2A1 and COL1A1 cartilage
			expression and promotes knee pathological changes.
			Promotes macrophage infiltration in the pannus,
MMP-12	[102,103,104]	AR	enhancing inflammation. Increases synovial
			thickening and cartilage destruction.
			Promotes inflammation and joint destruction in arthritis
	[105]	AR	models. Participates in cartilage destruction in modulating
			the inflammatory response.
MMP-13			Plays a crucial role in COL2, ADAMTS-4,
	[106,107]	OA	ADAMTS-5, and aggrecan degradation. Causes synovial
			hyperplasia and synovitis with diffuse mononuclear
			cell infiltration.
Chemokine			
	[108,109]	AR	Promotes the locomotion and recruitment of monocytes
CCL2 (MCP-1)			and macrophages to the inflamed joint.
	[110]	OA	Induces articular chondrocyte MMP-1, MMP-3, and
			MMP-13 upregulation.
	[111]	AR	Acts as a major chemoattractant for CCR5^+^ leucocytes
CCL4 (MIP-1β)			from peripheral blood to articular tissue.
	[112]	OA	Upregulated in response to IL-1β and resistin.
			Biomarker of K-OA severity.

Abbreviations: TNF-α: tumor necrosis factor alpha; RA: rheumatoid arthritis; OA: osteoarthritis; IL-6: interleukin-6; IL-1β: interleukin-1 beta; OC: osteoclast; ROS: reactive oxygen species; IL-17: interleukin 17; IL-18: interleukin 18; IL-21: interleukin 21; TGF-β: transforming growth factor beta; Treg: T regulatory cells; MMP: metalloproteinases; ADAMTS: A disintegrin and metalloproteinase with thrombospontin motifs; RANKL: receptor activator of nuclear factor kappa-B ligand; MSC: mesenchymal stem cells; FLS: fibroblast-like synoviocytes; JAK: Janus kinase; STAT: signal transducer and activator of transcription proteins; MAPK: mitogen-activated protein kinase; PI3K: phosphoinositide 3-kinase; Akt: protein kinase v-akt murine thymoma viral oncogene homolog; T_FH_: T follicular helper cell; T_H_17: T helper 17; COL2A1: collagen type 2 alpha 1 chain; COL1A1: collagen type 1 alpha 1 chain; COL2: collagen type 2; CCL2: C-C motif chemokine ligand 2; MCP-1: monocyte chemoattractant protein-1; CCL4: C-C motif chemokine ligand 4; MIP-1β: macrophage inflammatory protein 1 beta; CCR5: C-C chemokine receptor type 5; K-OA: knee osteoarthritis.

**Table 2 biomolecules-13-00502-t002:** Soluble osteopontin levels in plasma, serum, SF, and urine of rheumatoid arthritis and osteoarthritis patients compared to controls.

Sample	Ref.	*n*	sOPN Levels
Plasma	[118]	45 JIA	0.25 ng/mL (0–1.160 ng/mL)
	[66]	41 RA	9.93 ng/mL (4.36–47.80 ng/mL)
		28 SSc	4.3 ng/mL (2.1–19.7 ng/mL)
		18 Co	5.2 ng/mL (4.1–9.4 ng/mL)
	[119]	37 RA	RA-R = 34.8 ng/mL (39.8–1150.9 ng/mL)
			RA-NR = 114 ng/mL (38.4–854.9 ng/mL)
	[120]	70 RA	499.4 ng/mL (58.8–1492.8 ng/mL)
		20 OA	413.9 ng/mL (195.6–1047.8 ng/mL)
		46 Co	277.9 ng/mL (114.0–655.1 ng/mL)
	[121]	32 OA	K-OA = 168.8 ± 15.6 ng/mL
		15 Co	67.2 ± 7.7 ng/mL
			309.78 ng/mL (212.2–521.5 ng/mL)
	[122]	21 OA	H-OA KL-3 = 283.41 ng/mL (212.2–400.1 ng/mL)
			H-OA KL-4 = 362. 51 ng/mL (256.3–521.5 ng/mL)
			Pre-operative
			K-OA = 305.85 ± 55.31 ng/mL
	[123]	30 OA	H-OA = 303.06 ± ng/mL
			Post-operative
			K-OA = 434.2 ± 144.85 ng/mL
			H-OA = 376.14 ± 90.69 ng/mL
			Pre-operative
			155 ± 12.8 ng/mL
			Day 1 post-operative
	[124]	63 OA	152 ± 11.6 ng/mL
			Day 5–7 post-operative
			140 ± 13.8 ng/mL
			150 ± 7.20 ng/mL
		50 Co	66.3 ± 2.49 ng/mL
Serum			FL = 1.7 ± 1.1 ng/mL
	[125]	22 OA	N-half = 21.1 ± 11.7 ng/mL
			F-OA N-half = 23.2 ± 11.8 ng/mL
			M-OA N-half = 11.1 ± 4.9 ng/mL
		12 Co	FL = 1.8 ± 0.5 ng/mL
			N-half = 14.7 ± 3.6 ng/mL
			21.804 ± 14.932 ng/mL
			RA-M = 16.943 ± 8.055 ng/mL
	[126]	40 RA	RA-S = 25.397 ± 17.785 ng/mL
			RA-R baseline = 22.223 ± 20.106 ng/mL
			RA-R after DMARD = 11.248 ± 9.583 ng/mL
		11 Co	6.900 ± 4.593 ng/mL
			K-OA = 4.908 ± 0.769 ng/mL
	[127]	148 OA	KL-2 = 5.684 ± 0.727 ng/mL
			KL-3 = 4.494 ± 0.592 ng/mL
			KL-4 = 5.136 ± 0.673 ng/mL
		101 Co	2.182 ± 0.217 ng/mL
SF	[118]	45 JIA	2.55 ng/mL
	[121]	32 OA	K-OA = 272.1 ± 15.0 ng/mL
			K-OA = 4.519 ± 1.830 ng/mL
	[128]	50 OA	KL-2 = 3.543 ± 0.811 ng/mL
			KL-3 = 4.013 ± 0.676 ng/mL
			KL-4 = 6.170 ± 0.773 ng/mL
		10 Co	1.179. ± 0.303 ng/mL
	[125]	22 OA	FL = 7.75 ± 2.33 ng/mL
			F-OA N-half = 0.91 ± 0.49 ng/mL
			M-OA N-half = 0.12 ± 0.08 ng/mL
			KL-2 N-half = 0.669 ± 0.476 ng/mL
	[72]	750 OA	KL-3 N-half = 5.203 ± 0.385 ng/mL
			KL-4 N-half = 5.642 ± 0.350 ng/mL
	[129]	42 OA	3.378 ± 4.195 ng/mL
		40 Co	0.892 ± 0.323 ng/mL
Urine		70 RA	FL = 2044.6 ng/mmol Cr
			N-half = 143.5 pmol/mmol Cr
	[120]	20 OA	FL = 945.7 ng/mmol Cr
			N half = 69.8 pmol/mmol Cr
		46 Co	67.9 pmol/mmol Cr

Abbreviations: JIA: oligoarticular onset juvenile idiopathic arthritis; ng: nanograms; mL: milliliters; RA: rheumatoid arthritis; OA: osteoarthritis; sOPN: soluble osteopontin; Co: controls; SF: synovial fluid; SSc: systemic sclerosis; K-OA: knee osteoarthritis patients; K-OA SF: knee osteoarthritis patients’ synovial fluid; Cr: creatinine; FL: osteopontin full-length; N-half: thrombin-cleaved osteopontin; H-OA: hip osteoarthritis; RA-R: rheumatoid arthritis responders; RA-NR: rheumatoid arthritis non-responders; RA-M: rheumatoid arthritis patients with moderate disease activity evaluated by DAS28; RA-S: rheumatoid arthritis patients with severe disease activity evaluated by DAS28; DAS28: disease activity score in 28 joints; DMARD: disease-modifying antirheumatic drugs; F-OA: female osteoarthritis patients; M-OA: male osteoarthritis patients; KL: Kellgren–Lawrence grade.

**Table 3 biomolecules-13-00502-t003:** *SPP1* polymorphism performed in rheumatoid arthritis and osteoarthritis patients.

Population	Ref.	*n*	*SPP1* SNP	Main Findings
			327 T/C	
	[152]	263 RA	795 C/T	Strong linkage disequilibrium
		478 Co	1128 A/G	in *SPP1* SNP.
			1284 A/C	
			+245 first intron	Association of variant 2 with
	[154]	119 JIA	biallelic ins/del	persistent oligoarticular JIA vs.
		200 Co	(variant 1 = TG allele;	extended form.
			variant 2 = TGTG allele)	
				Association of rs11439060 with
European	[155]	377 RA	rs11439060 (−156/GG)	RA susceptibility (*p* = 0.023,
		391 Co	rs9138 (+1239 A/C)	OR 1.47).
				Association of rs9138 with
				joint destruction in 358 ACPA-
				negative patients (*p* = 0.034).
				Association of rs4754 in complete
	[156]	652 RA	rs9138	linkage disequilibrium with
		(ESPOIR)	rs11439060	rs9138 with joint damage
		273 RA	rs4754	progression in ACPA-negative
		(EAC)		patients at 2 (*p* = 0.019) and 7
				year follow-ups (*p* = 0.050).
			−631 G/T	
			−458 T/C	*SPP1* genotype and allele
		192 RA	rs4754	frequencies do not differ between
		288 Co	rs1126616	RA patients and Co.
			rs1126772	
			rs9138	
				Association of −443 C/T and −66
	[72]	750 OA	−156 GG/G	T/G with OA risk and
Han Chinese		794 Co	−443 C/T	radiographic severity, and
			−66 T/G	affect SF sOPN N-half levels.
		389 OA	rs17524488	rs17524488 delG > insG increase
	[158]	315 Co	rs11730582	H-OA risk (OR 1.48, 95% CI
				1.18–1.85, *p* < 0.01).
				rs11730582 C allele of C
		403 OA		genotype related to decreased
	[160]	536 Co	rs11730582	risk of H-OA. Association
				among females and BMI
				<25 kg/m^2^ groups.
Mexican	[159]	296 OA	rs11730582	No association between
		354 Co		rs11730582 and primary K-OA.
Nine			rs11439060	Association of rs11439060 and
independent	[157]	11,715 RA	(−156/GG)	rs9138 risk allele with ACPA
Cohorts *		26,493 C	rs9138	negative (*p* = 1.29 × 10^−5^).
			(+1239 A/C)	

*SPP1*: sialoprotein 1; SNP: single nucleotide polymorphism; sOPN: soluble osteopontin; RA: rheumatoid arthritis; Co: controls; FL: osteopontin full-length; N-half: thrombin-cleaved osteopontin; KL: Kellgren–Lawrence grade; JIA: oligoarticular onset juvenile idiopathic arthritis; ESPOIR: longitudinal prospective cohort of adults with possible early RA; EAC: Leiden early arthritis clinic cohort; ACPA: anti-citrullinated antibodies; H-OA: hip osteoarthritis; K-OA: knee osteoarthritis; OR: odds ratio; CI: confidence interval; BMI: body mass index.* French, Spanish, Swedish, and Japanese collections.

## Data Availability

Not applicable.

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
