# Peer review of "Osteopontin: A Bone-Derived Protein Involved in Rheumatoid Arthritis and Osteoarthritis Immunopathology"

_biomolecules, 2023, doi:10.3390/biom13030502_

Round 1

Reviewer 1 Report

Biomolecules

Review 1

Review biomolecules-2223474

Title :Osteopontin: A Bone-derived Protein involved in Rheumatoid Arthritis and Osteoarthritis Immunopathology

1.In this review  “section 2 - A brief look at the basic immunopathology of RA and OA”, can be curtailed to short, since matter is not much relevant to OPN

2.Avoid abbreviations in title/subtitle

3.Subtitle of section  1.1. “Osteopontin: a pleiotropic bone-derived protein”, is not appropriate since the matter belonging to this sector is reflecting various function of OPN and not specifically bone related function. Please justify.

4.As per the subtitle of sub section 1.3 –“OPN involvement in inflammatory diseases”, key role of OPN responsible in each  inflammatory disease  must be mentioned along with reference.

5.In section 2 there is negligible information of OPN. Please justify. The matter may be  reduced to relevant information.

6.In Section 3 author has written collective information of RA and OA. However, analysis of these information must be incorporated.

7. Unable to understand line 359 “we consider specify that the physiological sOPN  circulating levels range from 31 to 200 ng/mL”, reframe the sentence appropriatly.

8.What is constitutive SPP1/OPN and its mechanism related to RA? Please incorporate information.

9.Author provided collective information of various proteins, cytokines, their mechanism etc involved in either OA/RA/both cases. Comparative information in a table form along with role, reference would be more appropriate. It  must provided.

10.The title “regulation of OPN function in AR and OA” may be reframed since the content under this title is regarding micro RNA.

11. The collected information must be analysed and conclusion must be provided after each sub section.

11. Full form of all the  abbreviation  must be provided when appear 1st time in the manuscript. Ex:,  IKKb, p38, OPN N , AR, OPN-FL, p-Smad,NO, ACL, iNOS, PGE2, SPP1 SNP, NLRP3, ADANTS4, TN-C, OPN KO, sOPN etc.

Reviewer 2 Report

Minor comments for clarification and better reading of the tables:

Table 1+2: The columns "Author/year" and "Population" should be omitted to allow narrower lines and better overview of the table. Instead, the column "Ref" should be put first. In the main findings, "were" should be omitted and presentation shortened.

Line 201: The brief look should be really shortened to focus on the OPN-issue.

Table 1: In line 375, the authors present the table as "In Table 1 we summarize the main results 357 from studies performed in RA and OA patients in which sOPN levels were determined in 358 plasma, urine, and SF [53, 84, 88-92]". Accordingly the legend should be changed to "sOPN levels in plasma, urine, and synovial fluid of RA and OA patients compared to controls". Also, reference 84 (study by Ohsima et al.) should not be included as it stands for OPN-secretion, and the study by Tanaka et al (Ref. 95) can be mentioned in the text, but should not be included in this table. I propose "Co" instead "C" as abbreviation of Controls. 

Table 1, column "sOPN levels": All units should be the same (e.g. ng/ml). Also table 1 should be subdivided in plasma, synovial and urine studies for clarification and better comparisons of data. Maybe the authors then find a way to present weighted averages of OPN-levels in RA, OA and controls (plasma, synovial and urine) for better comparisons (if possible even before and after surgery). 

Table 2: Studies should be grouped according to "European", "Han-chinese", "Mexican" and others.

Tables should not be separated into 2 parts on following sides.

Line 650: RA instead of AR
